# Joint regression modeling of blood pressure and associated factors among adults in Uganda: Implications for clinical practice

**Saidi Appeli** [1] *, **Saint Kizito Omala** [2], **Jonathan Izudi** [3,4]

**1** Department of Agribusiness and Extension, Faculty of Agriculture and Animal Sciences, Busitema University, Soroti, Uganda, **2** Department of Statistical Methods and Actuarial Sciences, School of Statistics and Planning, Makerere University, Kampala, Uganda, **3** Department of Community Health, Faculty of Medicine, Mbarara University of Science and Technology, Mbarara, Uganda, **4** Data Synergy and Evaluations Unit, African Population and Health Research Center, Nairobi, Kenya

\* saidiappeli12@gmail.com

**Data Availability Statement:** We analyzed the 2014 Uganda STEPS NCD Risk Factors Survey data which is publicly available at

## Abstract

Blood pressure (BP) is a repeated measurement data as multiple measurements of both systolic blood pressure (SBP) and diastolic blood pressure (DBP) are simultaneously obtained on a patient to determine a raised blood pressure (hypertension). In examining factors associated with hypertension, BP is measured either as a binary outcome leading to information loss and reduced statistical efficiency or as a continuous outcome based on the average of one of the measurements or a combination of the two but independently thus ignoring possible correlation. We simultaneously modeled the risk factors for increased SBP and DBP among adults in Uganda and tested the difference in the effect of certain determinants on SBP versus DBP. We analyzed the 2014 nationwide non-communicable disease risk factor baseline survey data of Ugandans aged 18–69 years. We considered SBP and DBP as two continuous outcomes and conducted multivariate linear regression to jointly model SBP and DBP accounting for their distribution as bivariate normal. Of 3,646 participants, 950 (26.1%) had hypertension based on SBP (BP $\geq$ 140 mmHg) and DBP (BP $\geq$ 90 mmHg), 631 (17.3%) based on SBP alone, and 780 (21.4%) based on DBP alone. The study found that an increase in age (ranging from 18–69 years), obesity, income, being centrally obese, and hypercholesterolemia were significantly associated with higher SBP levels. Living in eastern, northern, and western Uganda regions was significantly associated with lower SBP, whereas increasing age, obesity, and hypercholesterolemia were significantly associated with higher DBP. Adults who rarely added salt to their meals were on average associated with higher DBP levels than those who never added salt to their meals. We found a strong residual correlation between SBP and DBP (r = 0.7307) even after accounting for covariates at the marginal level. This study presents a statistical technique for joint modeling of blood pressure, enabling the estimation of correlation between two outcomes and controlling family-wise error rate by testing the effect of a risk factor across both outcomes simultaneously.

https://ghdx.healthdata.org/record/uganda-steps-noncommunicable-disease-risk-factors-survey-2014. No ethical approval was needed as the data are de-identified and ethical approval was obtained in the past from Saint Francis Hospital, Nsambya Institutional Review Board in 2006 and renewed in 2013. The Uganda Ministry of Health NCDs Desk provided the dataset for this analysis following our request.

**Funding:** The authors received no specific funding for this work.

**Competing interests:** The authors have declared that no competing interests exist.

## Introduction

Raised blood pressure or hypertension is a worldwide public health challenge and a leading modifiable risk factor for cardiovascular diseases (CVDs), affecting approximately 1 billion people globally and accounting for over 7.8 million deaths annually [1]. Within sub-Saharan Africa (SSA), it is projected that about 125.5 million people will be affected by hypertension by 2025 [2]. Although blood pressure is a repeated measurement data, with multiple simultaneous measurements of both SBP and DBP on the same individual usually on a continuous scale, several studies [3–8] have analyzed blood pressure as either a binary outcome using the binary logistic regression model or as a continuous outcome using multiple linear regression model based on the average of one of the blood pressures or a combination of the two blood pressures. Although the binary logistic regression approach simplifies the statistical analysis, interpretation, and presentation of results, dichotomization of continuous measures like blood pressure leads to a significant loss of information about individual differences as it assumes individuals in the same stratum are homogeneous. The incomplete use of information from such measures leads to a reduction in precision and statistical efficiency in the identification of risk factors associated with raised blood pressure [9–11]. Conversely, using multiple linear regression based on the average of one of the blood pressure measurements or a combination of the two blood pressures as independent measurements ignores the association or correlation between SBP and DBP leading to biased results [12]. For example, leaving out a correlated outcome leads to omitted variable bias and failure to detect cross-effects between outcomes, overall biasing the regression coefficient estimates. Additionally, the model runs a high risk of producing spuriously significant results as separate Ordinary Least Squares (OLS) regression models neither produce multivariate results nor allow for testing of coefficients across the two equations.

Where the dependent variables are associated or correlated, multivariate analysis techniques for simultaneous analysis of data on correlated variables would be more appropriate and efficient than univariate techniques since additional information from the association can be used [12]. Currently, data regarding the application of the multivariate linear regression model in jointly modelling the correlates of SBP and DBP among adults in Uganda are scarce. Therefore, we simultaneously model raised SBP and DBP as bivariate and normally distributed data in assessing their risk factors among adults in Uganda. Additionally, we tested the differences in the effect of certain determinants on SBP versus DBP, which would have been impossible under separate OLS regression models. Our approach incorporates the association between the two blood pressures thus allowing the use of more information than the commonly used univariate analysis in order to better understand the strength and direction of the relationship.

Our study addressed a critical public health problem facing Ugandan adults from a methodological viewpoint demonstrating multivariate analysis techniques for simultaneous analysis of data on correlated dependent variables provide more appropriate and efficient results than the univariate techniques since additional information from the association is used. Our study underscores a need to use robust and more precise statistical approaches in identifying the risk factors for hypertension. Identification of risk factors with greater precision would contribute to improving risk stratification and can guide the development of public health prevention measures, including clinical care for people with hypertension.

## Materials and methods

### Description of the data source

The data analyzed in the present study were from the most recent (2014) NCD risk factor baseline survey conducted by the Uganda Ministry of Health (MoH).

The survey collected cross-sectional data using the World Health Organization (WHO) STEPwise approach to surveillance providing a standardized method for collecting, analyzing, and disseminating data on NCD risk factors. Data were collected in the WHO STEPs 1, 2, and 3 using personal digital assistants (PDAs). Socio-demographic and behavioral information were collected in STEP 1. Physical measurements were collected in STEP 2, and Biochemical measurements were collected in STEP 3. A multi-stage cluster sampling design was used to produce a nationwide representative sample in two stages. The first stage involved the selection of sample points or clusters from an updated master sampling frame constructed from the 2014 Uganda National Population and Housing Census (UNPHC) that employed a probability proportional to size approach. In the second stage, individuals were selected at the household level using a simple random sampling approach [13, 14]. The study population comprised adults aged 18–68 years who were residents in a particular cluster for at least six months in any of the four regions in Uganda: Central, Eastern, Northern, and Western. The survey had a sample of 3,987 respondents but this study was based on 3,646 adults with valid responses. The survey took three SBP and DBP measurements from an individual, measured 3–5 minutes apart on the left arm with the participant in the sitting position using a battery-powered digital blood pressure machine *(Boso Medicus Uno)* and recorded the measurements on a continuous scale in the NCD dataset. In each round of blood pressure measurements, the SBP and DBP were measured simultaneously by the machine.

## Ethical statement

We analyzed the 2014 Uganda STEPS NCD Risk Factors Survey data which is publicly available at https://extranet.who.int/ncdsmicrodata/index.php/catalog/633/.

No ethical approval was needed as the data were de-identified and ethical approval was obtained in the past from Saint Francis Hospital, Nsambya Institutional Review Board in 2006 and renewed in 2013. The Uganda Ministry of Health NCDs Desk provided the dataset for this analysis following our request.

## Variables and measurements

**Dependent variable.**    The current study considered systolic blood pressure and diastolic blood pressure as the two continuous outcome variables measured in units of millimeters of mercury (mmHg).

**Independent variables.**    The independent variables considered included the participants age measured on a continuous scale, sex (male or female), residence (urban or rural), level of education (no formal education, primary education, secondary education, and university or tertiary), income measured in the United States of America dollar or USD ($<277$, 278–2,777, and $\geq 2,778$), and marital status (unmarried, married or co-habiting, separated/widowed/divorced). Additional variables included the tobacco smoking status measured as do not smoke, smoke between 1–10, and smoke $> 10$; alcohol consumption measured on a binary scale (yes or no); and, the geographical region measured as central, eastern, northern, and western Uganda. Physical activity was measured on a binary scale (yes or no) based on whether or not an individual met the WHO recommendations on physical activity for health. Accordingly, adults who had at least 150 minutes of moderate-intensity physical activity, 75 minutes of vigorous-intensity physical activity, or an equivalent combination of moderate and vigorous-intensity physical activity achieving at least 600 Metabolic Equivalent of Task (MET)-minutes were considered physically active. MET-minutes is a unit of measurement used to quantify physical activity. It expresses the total amount of physical activity performed over time (energy expenditure of a specific activity), such as a day or a week.

One MET is equivalent to the energy expended at rest and is estimated to be about 1 kilocalorie (kcal)/kg/hour. Fasting blood glucose was initially measured as a continuous variable but later categorized as <6.1mmol/l, 6.1–6.9 mmol/l, and ≥7 mmol/l to depict normal fasting blood glucose, impaired, and diabetic status, respectively. Body mass index (BMI) was initially computed as a continuous variable by dividing the weight in kilograms (kg) by the height in meters of an individual, expressed as kilograms per squared meters (kg/m$^2$), and later categorized as <18.5kg/m$^2$, 18.5–24.9 kg/m$^2$, 25.0–29.9 kg/m$^2$, and ≥30 kg/m$^2$ to denote underweight, normal weight, overweight, and obese respectively. Total blood cholesterol was initially measured as a continuous variable and then categorized as <6.2mmol/L and ≥6.2mmol/L to depict normal (non-hypercholesterolemic) and high cholesterol levels (hypercholesterolemic), respectively. Central obesity was measured on a binary scale (obese or normal) based on the waist-hip-ratio (waist circumference/hip circumference) of >0.85 in women and >0.95 in men to depict central obesity else normal. During the survey, the waist and hip circumferences were measured to the nearest 0.5 cm using a non-stretchable standard tape measure [14]. The waist measurement was taken mid-way between the lowest rib and the iliac crest with the subject standing at the end of gentle expiration, and hip measurement at the greater trochanters. These variables were selected based on their biological relationship with hypertension from the literature. We excluded people on anti-hypertensive medication.

## Statistical analysis

Data management and analysis were conducted in Stata version 15.1 and R version 4.2.1 [15]. We descriptively summarized categorical data using frequencies and percentages and numerical data using the median, interquartile range (IQR) when skewed otherwise, the mean and standard deviation were used. To estimate the prevalence of hypertension, a value of 1 was assigned to average SBP ≥ 140 mmHg and average DBP ≥ 90 mmHg (hypertensive), and 0 was assigned to average SBP < 140 mmHg and average DBP < 90 mmHg (normotensive).

We examined the correlation between the dependent variables using a correlation matrix and a formal statistical test. The correlation between the two blood pressures was established by the significance of Pearson's test and justified the use of a multivariate linear regression model. To minimize the effect of multicollinearity, only predictor variables with variance inflation factor (VIF) less than 10 were included in the multivariable multivariate model. This was achieved by using the *"Collin"* command (a community-contributed package), a pre-estimation command that displays several different measures of multicollinearity including VIF, and only predictor (independent) variables are used. Furthermore, we performed variable predictor selection by fitting an exploratory multivariate linear regression model between each predictor variable and the outcomes (simultaneously) to establish their relationship and identify potential predictors. All the predictor variables with a probability value less than or equal to 0.20 in either of the response variable regression results from the exploratory multivariate linear regression model were considered for further investigation at the model estimation level. These predictor variables included age in years, income, marital status, geographical region, physical activity, eating at least five servings of fruits per day, fasting blood glucose, body mass index, dietary salt intake, central obesity, and total blood cholesterol. For the inferential analysis, we conducted two sets of models: modeling SBP and DBP separately in two separate linear regression models and a joint model for both SBP and DBP adjusted for the selected set of explanatory variables in identifying the correlates. The motivation for the separate linear models for SBP and DBP is to compare the results with that of the joint model, whereas the joint modeling of SBP and DBP was to address the correlation of the responses. The bivariate normal distribution model was chosen as several statistics such as sample mean, asymptotically

follow the normal distribution even if the original data do not follow normal distribution according to the central limit theorem [12]. Regression diagnostics were performed to assess the goodness of fit of the multivariate model to data. We assessed the joint significance of the two equations using the F-test to determine the overall effect of the predictor variables on the two responses.

The null hypothesis was that the predictor variables overall were not significantly associated with the two responses at a 5% significance level. A resulting $p<0.05$ led to the rejection of the null hypothesis. Bivariate normality of data assumption was evaluated using the *Shapiro–Wilk* test in the *mvnormtest package* by calling the function mshapiro.test, which is a generalization of the Shapiro–Wilk test for univariate normality. The null hypothesis was that data are bivariate normally distributed. Lastly, we tested for the independence of the residual correlation using the Breusch-Pagan test. The null hypothesis was that the residuals were not correlated. We present the results of these tests in the results section. Our analysis accounted for the complex survey design using parameters like primary sampling unit, strata, and weight since the NCD survey utilized a complex sample design. In this analysis, a predictor variable was reported as having a significant effect on the dependent variable if its effect on SBP or DBP was statistically significant at the 5% level of significance. The multivariate model specification equation is detailed below.

## Multivariate linear regression model specification in matrix form

The multivariate linear regression model in matrix form, according to Wu and Qiu [12], is described as follows:

Let $\mathbf{y}_i = (y_{i1}, y_{i2})^T$ be two continuous variables on individual i and let $\mathbf{z}_i = (z_{i1}, z_{i2}, \ldots, z_{ir})^T$ be r predictor variables on individual i, i = 1, 2,..., n. A multivariate regression model can be written as follows;

$$y_{i1} = \beta_{01} + \beta_{11}z_{i1} + \cdots + \beta_{r1}z_{ir} + \varepsilon_{i1}, \qquad (1)$$

$$y_{i2} = \beta_{02} + \beta_{12}z_{i1} + \cdots + \beta_{r2}z_{ir} + \varepsilon_{i2}$$

$$i = 1, 2, \ldots, n$$

where i is the number of observations, the vector of random errors $\varepsilon_i = (\varepsilon_{i1}, \varepsilon_{i2})^T$ is assumed to follow a multivariate normal distribution $\varepsilon_i \sim N_2(0, \Sigma)$. The covariance matrix $\Sigma$ measures the variances and correlations between the two response variables.

## Results

### Participant characteristics

Of 3,646 participants (Table 1), 2,178 (59.7%) were females, 2,616 (71.7%) lived in rural areas, 357 (9.8%) had attained university/tertiary education at the time of the survey, and 2,412 (66.2%) were married or cohabiting. The median age of the participants was 32 years (IQR: 25–44). Only 366 (10.0%) ate at least five servings of fruits daily and the corresponding number for servings of vegetables was 298 (8.2%). The majority (n = 3,390 or 93.0%) never smoked any tobacco products, and 1,742 (47.8%) consumed alcohol. Most (n = 3,029 or 83.1%) regularly engaged in physical activity and 236 (6.5%) had obesity. The hypertension prevalences were 950 (26.1%) based on SBP or DBP, 631 (17.3%) based on SBP alone, and 780 (21.4%) based on DBP alone. 1,166 (31.9%) participants had central obesity.

**Table 1. Description of the characteristics of the participants.**

| Characteristics | Levels | No. (%) |
|---|---|---|
| Age (years) | Median (IQR) | 32 (25–44) |
| Sex | Male | 1468 (40.3) |
| | Female | 2178 (59.7) |
| Residence | Urban | 1030 (28.3) |
| | Rural | 2616 (71.7) |
| Level of education | No formal education | 577 (15.8) |
| | Primary education | 1479 (40.6) |
| | Secondary education | 1233 (33.8) |
| | University or tertiary | 357 (9.8) |
| Region | Central | 1294 (35.5) |
| | Eastern | 964 (26.4) |
| | Northern | 779 (21.4) |
| | Western | 609 (16.7) |
| Total blood cholesterol | Hypercholerolemic | 313 (8.6) |
| | Not hypercholesterolemic | 3333 (91.4) |
| Income (USD) | <277 | 116 (3.2) |
| | 278–2,777 | 1951 (53.5) |
| | ≥2778 | 1579 (43.3) |
| Marital status | Never married | 583 (16.0) |
| | Married or co-habiting | 2412 (66.2) |
| | Separated/widowed/divorced | 651 (17.8) |
| A daily serving of fruits | Yes | 366 (10.0) |
| | No | 3280 (90.0) |
| Dietary salt intake | Rarely or never | 1939 (53.2) |
| | Always | 435 (12.0) |
| | Often/ or sometimes | 1272 (34.8) |
| Daily servings of vegetables | Yes | 298 (8.2) |
| | No | 3348 (92.8) |
| Smoking status | Never | 3390 (93.0) |
| | Smoked between 1–10 tobacco products | 213 (5.8) |
| | Smoked > 10 tobacco products | 43 (1.2) |
| Alcohol drinking | Yes | 1742 (47.8) |
| | No | 1904 (52.2) |
| Physical activity | Yes | 3029 (83.1) |
| | No | 617 (16.9) |
| Fasting blood glucose (mmol/l) | Normal | 3553 (97.5) |
| | Impaired | 67(1.8) |
| | Diabetic | 26 (1) |
| Body mass index (kg/m$^2$) | Underweight | 319 (8.8) |
| | Normal weight | 2506 (68.7) |
| | Overweight | 585 (16.0) |
| | Obese | 236 (6.5) |
| Hypertensive based on only SBP ≥140 mm/Hg | No | 3015 (82.7) |
| | Yes | 631 (17.3) |
| Hypertensive based on only DBP ≥90 mm/Hg | No | 2866 (78.6) |
| | Yes | 780 (21.4) |

(*Continued*)

**Table 1.** (Continued)

| Characteristics | Levels | No. (%) |
|---|---|---|
| Hypertensive based on both SBP (≥140 mm/Hg) or DBP (≥90 mm/Hg) | No | 2696 (73.9) |
| | Yes | 950 (26.1) |
| Central obesity | Normal | 2480 (68.1) |
| | Obese | 1166 (31.9) |

IQR = Interquartile range; USD: United States Dollar.

## Factors associated with SBP and DBP

**Results of SBP and DBP for separate univariate response models.** Table 2 presents the results of both the separate univariate linear regression and the joint model of the two blood pressure measurements (SBP or DBP). The aim was to compare the results of the univariate models with the results of the joint model. In the separate univariate models, body mass index, age, and total blood cholesterol level were statistically significantly associated with higher SBP and higher DBP. Region of residence, central obesity, and income level were significantly associated with SBP alone, while dietary salt intake was associated with DBP only.

**Results of joint multivariate response models.** First, we examined the correlation between SBP and DBP and tested its significance. We found SBP and BP were correlated (r = 0.7307, p<0.001) hence justifying a need for multivariate regression modeling. Second, the F-test for the joint significance of the two equations was conducted and the significance of the test suggested that the set of predictor variables as a whole was strongly significant (F (38, 3619) = 4.20), p< 0.001). Third, the assumption of the bivariate normality was tested, and the result indicated no statistical significance (p > 0.05), implying that the assumption was not violated.

Lastly, the Breusch-Pagan's (chi2(1) = 1942.705, Prob > chi2 = < 0.001) test of independence of the residual correlations from the two models indicated that the residuals were correlated. These findings showed the appropriateness of using the multivariate linear regression model in explaining the data.

The results of the joint multivariate response regression model are also presented in Table 2. Overall, the results are similar to that of the univariate regression model, except for the quantification of the estimated residual correlation between SBP and DBP (σ = 0.7307) even after adjusting for the independent variables which is not possible under the separate univariate model results.

**Results of SBP from the joint response model.** Every 1-year increase in age (range, 18–69 years), obesity, income levels of at least 2,778 USD per annum, being centrally obese, and hypercholesterolemia were significantly associated with higher SBP whereas residing in the eastern, northern, and or western compared to the central region were significantly associated with a lower SBP.

**Results of DBP from joint response model.** Every 1-year increase in age (range, 18–69 years), obesity compared to normal weight, and hypercholesterolemia were significantly associated with higher DBP. Individuals who rarely added salt to their meals (either before or when eating the meal) on average were associated with higher levels of DBP compared to those who never added salt to their meals.

In Table 3, we tested the hypothesis that the effect of a risk factor on SBP is the same as on the DBP for the significant factors in the joint multivariate response model. Our results showed that the effects of age of the adults, body mass index, geographical region of residence,

**Table 2. Results of independent univariate and joint multivariate response models of SBP and DBP.**

| Characteristics | Independent univariate models of SBP and DBP | | | | | | Joint multivariate response models of SBP and DBP | | | | | |
|---|---|---|---|---|---|---|---|---|---|---|---|---|
| | SBP | | | DBP | | | SBP | | | DBP | | |
| | Coef | SE | (95% CI) | Coef | SE | (95% CI) | Coef | SE | (95% CI) | Coef | SE | (95% CI) |
| **Age (years)** | 0.22*** | 0.02 | (0.19, 0.26) | 0.24*** | 0.08 | (0.08, 0.40) | 0.22*** | 0.02 | (0.19, 0.25) | 0.24*** | 0.07 | (0.08, 0.40) |
| **Body mass index (kg/m$^2$)** | | | | | | | | | | | | |
| Normal weight | Ref | | | | | | | | | | | |
| Underweight | -1.89* | 1.14 | (-4.13, 0.34) | -0.34 | 0.73 | (-1.77, 1.09) | -1.89* | 1.14 | (-4.13, 0.34) | -0.34 | 0.73 | (-1.77, 1.09) |
| Overweight | 0.38 | 0.89 | (-1.37, 2.13) | 0.54 | 0.57 | (-0.58, 1.66) | 0.38 | 0.89 | (-1.37, 2.12) | 0.54 | 0.57 | (-0.58, 1.65) |
| Obese | 4.00*** | 1.31 | (1.42, 6.57) | 2.53*** | 0.84 | (0.88, 4.18) | 4.00*** | 1.30 | (1.41, 6.57) | 2.53*** | 0.84 | (0.88, 4.18) |
| **Fasting blood glucose (mmol/l)** | | | | | | | | | | | | |
| Normal | Ref | | | | | | | | | | | |
| Impaired | 3.69 | 2.31 | (-0.85, 8.22) | 0.68 | 1.48 | (-2.23, 3.58) | 3.69 | 2.30 | (-0.84, 8.22) | 0.68 | 1.48 | (-2.23, 3.58) |
| Diabetic | 1.95 | 3.54 | (-4.99, 8.90) | 0.18 | 2.27 | (-4.27, 4.63) | 1.95 | 3.54 | (-5.00, 8.90) | 0.18 | 2.26 | (-4.27, 4.62) |
| **Region** | | | | | | | | | | | | |
| Central | Ref | | | | | | | | | | | |
| Eastern | -2.95*** | 0.91 | (-4.74, -1.16) | 0.27 | 0.59 | (-0.87, 1.42) | -2.95*** | 0.90 | (-4.74, -1.16) | 0.27 | 0.58 | (-0.87, 1.42) |
| Northern | -4.84*** | 1.01 | (-6.83, -2.85) | -0.21 | 0.65 | (-1.48, 1.07) | -4.84*** | 1.01 | (-6.82, -2.85) | -0.21 | 0.64 | (-1.47, 1.06) |
| Western | -5.26*** | 0.93 | (-7.09, -3.43) | -0.23 | 0.60 | (-1.40, 0.94) | -5.26*** | 0.93 | (-7.08, -3.43) | -0.23 | 0.59 | (-1.40, 0.94) |
| **Marital status** | | | | | | | | | | | | |
| Never married | Ref | | | | | | | | | | | |
| Married/cohabiting | 0.94 | 0.95 | (-0.92, 2.79) | 0.02 | 0.61 | (-1.16, 1.21) | 0.94 | 0.94 | (-0.91, 2.79) | 0.02 | 0.60 | (-1.16, 1.21) |
| Separated/widowed | -0.95 | 1.24 | (-3.38, 1.48) | -1.45* | 0.79 | (-3.01, 0.10) | -0.95 | 1.23 | (-3.38, 1.47) | -1.45* | 0.79 | (-3.00, 0.10) |
| **A daily serving of fruits** | | | | | | | | | | | | |
| No | Ref | | | | | | | | | | | |
| Yes | -1.66 | 1.09 | (-3.80, 0.48) | -0.91 | 0.70 | (-2.27, 0.46) | -1.66 | 1.09 | (-3.80, 0.47) | -0.91 | 0.69 | (-2.27, 0.46) |
| **Dietary salt intake** | | | | | | | | | | | | |
| Always | Ref | | | | | | | | | | | |
| Often/sometimes | 1.41 | 1.06 | (-0.66, 3.49) | 0.97 | 0.68 | (-0.36, 2.30) | 1.41 | 1.05 | (-0.65, 3.49) | 0.97 | 0.67 | (-0.36, 2.29) |
| Rarely/never | 1.83* | 1.01 | (-0.16, 3.81) | 1.27** | 0.65 | (0.03, 2.54) | 1.83* | 1.01 | (-0.16, 3.80) | 1.27** | 0.64 | (0.03, 2.54) |
| **Physical activity** | | | | | | | | | | | | |
| No | Ref | | | | | | | | | | | |
| Yes | -0.71 | 0.86 | (-2.40, 0.97) | 0.10 | 0.55 | (-0.98, 1.17) | -0.71 | 0.85 | (-2.39, 0.97) | 0.10 | 0.55 | (-0.98, 1.17) |
| **Income (USD)** | | | | | | | | | | | | |
| < 277 | Ref | | | | | | | | | | | |
| 278–2,777 | -3.24* | 1.80 | (-6.77, 0.29) | -1.44 | 1.15 | (-3.70, 0.82) | -3.24* | 1.79 | (-6.76, 0.29) | -1.44 | 1.15 | (-3.69, 0.82) |
| ≥2,778 | 2.09*** | 0.66 | (0.79, 3.39) | 0.79* | 0.42 | (-0.04, 1.62) | 2.09*** | 0.66 | (0.79, 3.39) | 0.79* | 0.42 | (-0.04, 1.62) |
| **Total blood cholesterol** | | | | | | | | | | | | |
| Not hypercholesterolemic | Ref | | | | | | | | | | | |
| Hypercholesterolemic | 11.09*** | 2.41 | (6.36, 15.83) | 6.89*** | 1.55 | (3.86, 9.93) | 11.09*** | 2.40 | (6.36, 15.82) | 6.89*** | 1.54 | (3.86, 9.92) |
| **Central obesity** | | | | | | | | | | | | |
| Normal | Ref | | | | | | | | | | | |
| Obese | 1.98*** | 0.75 | (0.52, 3.45) | 0.71 | 0.48 | (-0.23, 1.65) | 1.98*** | 0.74 | (0.52, 3.44) | 0.71 | 0.47 | (-0.23, 1.64) |
| **Constant** | 122.98*** | 1.53 | (119.97, 125.98) | 78.84*** | 0.98 | (76.92, 80.77) | 122.98*** | 1.52 | (119.97, 125.98) | 78.84*** | 0.98 | (76.92, 80.77) |
| Sample | 3,646 | | | | | | 3,646 | | | | | |
| F (38, 3,619) | | | | | | | 4.20 | | | | | |
| Prob > F | | | | | | | <0.001 | | | | | |
| Residual correlation estimate | | | | | | | 0.7307 | | | | | |

*(Continued)*

**Table 2.** (Continued)

| Characteristics | Independent univariate models of SBP and DBP | | Joint multivariate response models of SBP and DBP | |
|---|---|---|---|---|
| | SBP | DBP | SBP | DBP |
| Breusch-Pagan test of independence: Chi-square (p-value) | | | 1942.705 (<0.001) | |

**Note**:

Statistical significance at 5% level:

*** p<0.01,

** p<0.05,

* p<0.1;

Ref: Reference category adopted in the regression; Coef: beta coefficients; CI: Confidence interval. SE: Standard errors

income, total blood cholesterol, and central obesity on SBP were significantly different from that of DBP, however, no such differences exist for dietary salt intake.

## Discussion

In this study, we jointly and separately modeled SBP and DBP in assessing their risk factors among adults in Uganda and tested the difference in the effect of some correlates on SBP versus DBP, which is impossible under separate OLS regressions. This approach allows the use of more information than the commonly used univariate analysis in order to better understand the strength and direction of the relationship. This approach of joint modeling of the risk factors of SBP and DBP is consistent with previous literature that argues the Joint statistical modeling of correlated outcomes addresses the correlation between the responses [16–20].

Based on the joint multivariate regression models for SBP and DBP, we identified age, body mass index, and total blood cholesterol level as significantly associated with SBP and DBP. Specifically, on average, every extra age was positively associated with higher blood pressure levels. Our finding of an association between age as a risk factor for SBP and DBP among adults in Uganda is similar to findings in past studies that also indicate blood pressure is associated with increasing age [2, 21–23]. The direct relationship between age and blood pressure among adults in Uganda in this study could be attributed to aging which results from changes in stiffness of the large arteries hence increased peripheral vascular resistance resulting in increased blood pressure. Furthermore, hypercholesterolemic individuals on average were significantly associated with higher levels of blood pressure compared to non-hypercholesterolemic individuals. This finding is consistent with a past study that reported a high prevalence of hypertension among hypercholesterolemic individuals [24]. The positive effect of hypercholesterolemia on blood pressure could in part be due to familial hypercholesterolemia and increased intake of saturated animal fats reported in many counties in sub-Saharan Africa [25]. Hypercholesterolemia is known to lead to the buildup of fats in the blood vessels hence narrowing them and causing a rise in blood pressure [26].

Consistent with previous studies that reported a higher prevalence of hypertension among the obese [27–30], obesity in this study was on average positively and significantly associated with higher SBP and DBP compared to normal weight. Compared with normal-weight adults in Uganda, the risk of blood pressure was not statistically significantly different for underweight and overweight adults. Body mass index is majorly influenced by dietary practices like high intake of saturated fatty acids and physical inactivity among others. A possible

**Table 3. Test for differences in the effect of risk factors across SBP and DBP.**

| Risk factors | Degree of freedom | F test value | p-value |
| --- | --- | --- | --- |
| **Age (years)** | 1 | 7.62 | <0.001*** |
| **Body mass index (kg/m$^2$)** | | | |
| Normal weight (Ref) | | | |
| Underweight | 1 | 3.90 | 0.048** |
| Overweight | 1 | 0.07 | 0.795 |
| Obese | 1 | 2.62 | 0.106 |
| **Region** | | | |
| Central (Ref) | | | |
| Eastern | 1 | 26.22 | <0.001*** |
| Northern | 1 | 44.03 | <0.001*** |
| Western | 1 | 61.15 | <0.001*** |
| **Dietary salt intake** | | | |
| Always (Ref) | | | |
| Often/sometimes | 1 | 0.37 | 0.541 |
| Rarely/never | 1 | 0.63 | 0.423 |
| **Income (USD)** | | | |
| < 277(Ref) | | | |
| 278–2,777 | 1 | 2.11 | 0.146 |
| ≥2,778 | 1 | 8.07 | 0.005*** |
| **Total blood cholesterol** | | | |
| Not hypercholesterolemic (Ref) | | | |
| Hypercholesterolemic | 1 | 6.38 | 0.011** |
| **Central obesity** | | | |
| Normal (Ref) | | | |
| Obese | 1 | 6.14 | 0.013** |

**Note**:

Statistical significance at 5% level:

*** p<0.01,

** p<0.05,

* p<0.1;

Ref: Reference category adopted in the tests; Ref: Reference category.

explanation for the higher levels of blood pressure among obese adults in Uganda could include a combination of factors like physical inactivity, sedentary lifestyles, and unhealthy diets (especially in urban settings) with minimal diversity.

We found individuals residing in the eastern, northern, and western regions of Uganda as compared to the Central region were significantly associated with lower SBP levels contrary to the findings by Guwatudde et al. [31] who reported no significant effect on residence on hypertension. This inconsistency could be attributed to differences in the modelling approach of the dependent variables, with continuous outcomes in the present study compared to dichotomous outcomes created from the two blood pressure measures in the past study. However, another probable reason is that Central Uganda is predominantly urban, with many individuals having better income levels, adopting a more sedentary lifestyle, and being exposed to unhealthy diets (junk foods, little dietary diversity, and fewer fruits and vegetables). Indeed, having an income level of nearly 3,000 USD per annum was, on average, significantly associated with higher SBP levels. Anjulo et al. [4] in Ethiopia found that income had no significant

effect on high blood pressure among adults, which is inconsistent with our results. Our finding of an association between income and higher SBP levels might be that people with high-income levels live sedentary or affluent lifestyles, are physically inactive, and get exposed to junk foods as stated earlier.

We found that participants who had central obesity significantly had higher levels of SBP compared to those without central obesity. The findings from this study are consistent with results from a past study where central obesity was found to be significantly associated with higher odds of high blood pressure [23, 32]. Central obesity is accompanied by insulin resistance, which potentially increases blood pressure [32].

Our finding that on average, individuals who rarely added salt to their meals either before or when eating were associated with higher levels of DBP than those who never added salt to the meal is consistent with the findings of Ware et al. [33] in South Africa. While there is a general understanding that excessive salt intake can contribute to the development of hypertension in some individuals, because salt is known to increase vascular osmotic pressure, attracting more fluids into the blood vessel and hence raising blood pressure, the relationship is not entirely straightforward. For example, although our study was among adults, in young subjects, salt intake is reported to have a "J-shaped relationship" with higher levels of blood pressure, blood pressure remains elevated even after a high salt intake has been reduced [34]. The findings also revealed that the effect of age of the adults, body mass index, geographical region of residence, income, total blood cholesterol, and central obesity on SBP differ significantly across SBP and DBP outcomes.

## Methodological and statistical implications of our findings

We finalize this discussion by highlighting the advantages of the methodological and statistical approach used in the present study as compared to previous studies. Our statistical analysis approach allows the correlation between two responses to be estimated while adjusting for the effects of independent variables. The results show that the two blood pressure measurements (SBP and DBP) are strongly and positively correlated with each other even after adjusting for the independent variables at the joint multivariate level analysis.

Furthermore, the multivariate/joint response modeling approach reduces the occurrence of type I error (the probability of rejecting the null hypothesis that is true in the population) and therefore produces estimates that are highly accurate and reliable compared to modeling such outcomes as separate entities, especially when they are correlated.

## Strengths and limitations of this study

Our study has strengths and limitations. To the best of our knowledge, this is the first study in Uganda to jointly model SBP and DBP among adults in Uganda in assessing their factors. Our analysis used a nationally representative sample so the findings are likely generalizable to the entire country and other similar settings. Limitations include a possibility of social desirability bias during self-reported responses, and a potential measurement bias during blood pressure takings, readings, or both. The dataset analyzed had missed some important family-level covariates. For example, the family history of hypertension of an individual. Additionally, the type of alcohol or alcohol content with potential differentials in effect and the quantity consumed were not investigated. Lastly, this study was unable to examine the changes that may have occurred in the adult population over time as the nationwide non-communicable disease baseline survey was conducted nearly a decade ago.

## Conclusion

This study has illustrated the joint modelling of blood pressure and its associated factors is a more efficient and easily applicable statistical technique that allows for the estimation of the correlation between the two responses while adjusting for the effects of the independent variables. Compared to the naïve separate univariate models, this approach controls the family-wise error rate by simultaneously testing the effect of a given risk factor across both outcomes.

Body mass index, total blood cholesterol, and central obesity are modifiable risk factors for raised blood pressure that can be addressed through lifestyle interventions. More research needs to be done to find the pathways through which age influences raised blood pressure. Similarly, interactions between and among the covariates included in this study should be investigated in future studies.

## Acknowledgments

The authors would like to thank the Uganda Ministry of Health, Non-Communicable Diseases (NCDs) Desk for unconditionally supporting this study by granting data access and providing administrative clearance.

## Author Contributions

**Conceptualization:** Saidi Appeli, Saint Kizito Omala, Jonathan Izudi.

**Data curation:** Saidi Appeli, Jonathan Izudi.

**Formal analysis:** Saidi Appeli, Jonathan Izudi.

**Methodology:** Saidi Appeli, Saint Kizito Omala, Jonathan Izudi.

**Software:** Saidi Appeli.

**Supervision:** Saint Kizito Omala, Jonathan Izudi.

**Writing – original draft:** Saidi Appeli, Saint Kizito Omala, Jonathan Izudi.

**Writing – review & editing:** Saint Kizito Omala, Jonathan Izudi.

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
