## [Decision Letter · Decision Letter 0]

30 May 2024

PGPH-D-24-00125

Joint regression modelling of blood pressure and associated factors among adults in Uganda: implications for clinical practice

Dear Dr. Appeli,

Thank you for submitting your manuscript to PLOS Global Public Health. After careful consideration, we feel that it has merit but does not fully meet PLOS Global Public Health’s publication criteria as it currently stands. Therefore, we invite you to submit a revised version of the manuscript that addresses the points raised during the review process.

The manuscript has been evaluated by two reviewers, and their comments are available below.

The reviewers raised a number of concerns. Specifically, they recommend that you expand the introduction section, that you improve the methodological and statistical analysis and that you reorganize the discussion section. Could you please carefully revise the manuscript to address all comments raised? 

We look forward to receiving your revised manuscript.

Kind regards,

Johanna Pruller, Ph.D.

PLOS Staff Editor

Journal Requirements:

Additional Editor Comments (if provided):

Reviewers' comments:

Reviewer's Responses to Questions

**Comments to the Author**

1. Does this manuscript meet PLOS Global Public Health’s publication criteria? Is the manuscript technically sound, and do the data support the conclusions? The manuscript must describe methodologically and ethically rigorous research with conclusions that are appropriately drawn based on the data presented.

Reviewer #1: Yes

Reviewer #2: No

2. Has the statistical analysis been performed appropriately and rigorously?

Reviewer #1: Yes

Reviewer #2: Yes

3. Have the authors made all data underlying the findings in their manuscript fully available (please refer to the Data Availability Statement at the start of the manuscript PDF file)?

Reviewer #1: Yes

Reviewer #2: Yes

4. Is the manuscript presented in an intelligible fashion and written in standard English?

Reviewer #1: Yes

Reviewer #2: No

5. Review Comments to the Author

Reviewer #1: The paper aims to contribute to the field of epidemiology by examining the sociodemocratic and lifestyle factors associated with systolic and diastolic blood pressure among adults in Uganda using a multivariate analysis approach. The use of a large, nationally representative sample from the 2014 Uganda STEPS NCD Risk Factors Survey and the employment of a multi-stage cluster sampling design provide a solid empirical basis for the study. The paper’s originality is evidenced in its focus on a Ugandan population, which adds to the body of knowledge regarding blood pressure variability in a lesser-studied African context. Furthermore, the application of a multivariate linear regression model to identify correlates of systolic and diastolic blood pressure simultaneously is a methodological strength that could yield insights into the multifactorial nature of hypertension.

The paper’s methodology section is a particular strength, as it outlines a robust statistical approach to data analysis including the use of primary sampling unit, strata, and weight adjustments for the complex survey design. Another strength is the relevance of the study to public health, given the growing burden of non-communicable diseases in low- and middle-income countries. The thorough investigation into a wide range of independent variables, including behavioral and metabolic factors, and their relationship with blood pressure, is commendable.

Reviewer #2: Overall impression

This is a technically well conducted study, but it is unclear what it adds to the scientific literature and does not seem to hold much significance. The authors state multivariate regressions are better/more efficient than univariate, but their own results do not bear this out. In addition, a more robust approach would have been to use simulations to show the difference in precision of estimates between univariate vs multivariate across multiple "datasets", rather than just applying it once to one dataset.

MAJOR ISSUES

Introduction

• Line 58: what kind of biased results? Need more detail here. “Conversely, using multiple linear regression based on the average of one of the blood pressure measurements or a combination of the two blood pressures as independent measurements ignores the association or correlation between SBP and DBP leading to biased results [12].”

• Line 69: but what is the benefit of “use of more information”? What additional information is gained from simultaneously modelling raised SBP and DBP, than in univariate analysis?

o How is this more efficient? Need to quantify this statement

Methods

• Line 115: It’s not clear how the two separate variables SBP and DBP were combined together into one outcome, as described in the Abstract. As it reads, it sounds like the authors kept the SBP and DBP as separate outcome variables

• Line 133: why were people with hypertension excluded? Only those on hypertension medications should be excluded

• Line 141: how was multicollinearity checked? Include details

• Line 144: there are major problems with variable selection using this method. https://journalofbigdata.springeropen.com/articles/10.1186/s40537-018-0143-6. I'd focus on the multivariable regression model (without selecting covariates stepwise, but examining collinearity) and the size of estimated associations, with p-values being secondary.

• Line 175: this section is dense and difficult to follow as a clinical researcher (ie not a statistician). It will be hard to approach for a general audience like PLOS Global Health. What does the matrix mean? What commands did you use in Stata to achieve this. What are the implications of using a matrix, how does it account for the correlation between the response variables?

Results

• Line 189-191: you said in Methods you excluded people with hypertension. So why would you be presenting the precent of people with hypertension here? It should be 0 if you excluded them

• Line 203-212: none of these statistical tests to assess assumptions were explained in Methods. Everything detailed in Results should be mentioned in Methods

• There is nothing here to show/justify why a multivariate model is more efficient/precise than univariate model. Need to compare the results from a multivariate vs univariate model, and show exactly how the multivariate model is “better”

o In fact, authors state “Overall, the results are similar to that of the univariate regression model”, so haven’t they proven that the multivariate model is not in fact better?

Discussion

• Organization here is lacking. First paragraph should focus on this study’s results. Comparison with other studies should be separate paragraphs.

• This focus is all on the risk factors for hypertension, but this is not the point of the study as outlined in the Introduction. The point is that multivariate analysis is better/more efficient than univariate for correlated dependent variables. Thus the discussion should focus on the latter point, and bring in prior statistical work on multivariate vs univariate analyses

• Line 287: “The implication of such a positive correlation between SBP and DBP is that either interventions designed to reduce the prevalence of hypertension based on SBP will also reduce the prevalence of hypertension based on DBP.” This is obvious and already known in routine clinical practice. I don’t think this study added much to that understanding

• Line 290: “Furthermore, compared to the naïve separate univariate models, our approach helps control the family-wise error rate - the probability of rejecting at least one of the null hypotheses for a given covariate by testing the effect of a given risk factor across both outcomes simultaneously.” As a non-statistician, I could not understand this sentence. What does “control the family-wise error rate” mean, does multivariate analysis reduce this error rate, or eliminate it? Can you give an example to make this concrete?

• Line 293: “Disregarding this model estimation aspect, as in the fitting of separate regression models for each blood pressure, could lead to imprecision in parameter estimates hencing biasing the results.” How would it bias results? Higher/lower? Also, your results ddn’t show this. Your multivariate models had almost the exact same 95% CI as the univariate, so they were not more precise.

MINOR ISSUES

Abstract

• Used cross-sectional data, cannot use terms like “led to decreased SBP”. Decrease in SBP implies SBP changing in same person over time. Should use “associated with lower SBP”. Same for “increased DBP”

Introduction

Methods

• Lines 110-113: this is redundant with text above. Would cut

• Line 116: typo, “and the and these”

• Lines 125-131: need to include details of how these variables defined. For example, for physical activity, levels for fasting blood glucose, BMI, etc

• Line 134: typo “individual”

Results

• Table 1: hypertension is defined clinically as either SBP >=140 or DBP >= 90. Rather than have separate rows for SBP Hypertensive, and DBP Hypertensive, I’d suggest re-organizing this so it’s Hypertension (yes/no), and then under hypertension have SBP >= 140, DBP >=80, or both as three separate rows

• Line 219: Used cross-sectional data, cannot use terms like “increased SBP” as this implies repeated measurements of same individual over time.

6. PLOS authors have the option to publish the peer review history of their article (what does this mean?). If published, this will include your full peer review and any attached files.

**Do you want your identity to be public for this peer review?** For information about this choice, including consent withdrawal, please see our Privacy Policy.

Reviewer #1: **Yes: **Olivier Mukuku

Reviewer #2: No

---

## [Decision Letter · Decision Letter 1]

19 Jul 2024

PGPH-D-24-00125R1

Joint regression modelling of blood pressure and associated factors among adults in Uganda: implications for clinical practice

Dear Dr. Appeli,

Thank you for submitting your manuscript to PLOS Global Public Health. After careful consideration, we feel that it has merit but does not fully meet PLOS Global Public Health’s publication criteria as it currently stands. Therefore, we invite you to submit a revised version of the manuscript that addresses the points raised during the review process.

Your manuscript has been evaluated by one of the previous reviewers, and one new reviewer (Reviewer 3); their comments are appended below.

While Reviewer 1 is positive towards publication, Reviewer 2 has identified some concerns regarding the study design, particularly regarding inclusion of hypothesis-driven research questions, among other tests and explanations noted in their report. Please ensure you address each of the reviewer's comments when revising your manuscript.

We look forward to receiving your revised manuscript.

Kind regards,

Hugh Cowley

Staff Editor

Additional Editor Comments (if provided):

Reviewers' comments:

Reviewer's Responses to Questions

**Comments to the Author**

1. If the authors have adequately addressed your comments raised in a previous round of review and you feel that this manuscript is now acceptable for publication, you may indicate that here to bypass the “Comments to the Author” section, enter your conflict of interest statement in the “Confidential to Editor” section, and submit your "Accept" recommendation.

Reviewer #1: All comments have been addressed

Reviewer #3: (No Response)

2. Does this manuscript meet PLOS Global Public Health’s publication criteria? Is the manuscript technically sound, and do the data support the conclusions? The manuscript must describe methodologically and ethically rigorous research with conclusions that are appropriately drawn based on the data presented.

Reviewer #1: Yes

Reviewer #3: Yes

3. Has the statistical analysis been performed appropriately and rigorously?

Reviewer #1: Yes

Reviewer #3: Yes

4. Have the authors made all data underlying the findings in their manuscript fully available (please refer to the Data Availability Statement at the start of the manuscript PDF file)?

Reviewer #1: Yes

Reviewer #3: Yes

5. Is the manuscript presented in an intelligible fashion and written in standard English?

Reviewer #1: Yes

Reviewer #3: Yes

6. Review Comments to the Author

Reviewer #1: (No Response)

Reviewer #3: The paper examined determinants of SBP and DBP in Uganda using multivariate multiple regression, accounting for the potential correlation between the two dependent variables. This is the first time I have reviewed the paper and I have focussed mainly on the current version of the paper.

Major comments:

1. The motivation for using the specific regression technic is sound and implementation appropriate. However, the results were presented mainly as a comparison with separate OLS regressions which is a trivial point because the coefficients should be identical. Instead, the authors should look into more hypothesis-driven research questions, for example to test the difference in the effect of certain determinants on SBP versus DBP, which wouldn't have been possible by using separate OLS regressions.

2. I suggest the authors to consider including waist circumference or waist-to-height ratio as an independent variable.

3. Given that DBP tends to decrease in older ages, the author should test the linearity of age effect and potentially include age as either a categorical variable or include a non-linear function of age.

4. The authors should explain how interactions between independent variables were considered.

5. Given that in theory separate vs joint modelling should give the same results, including SE, I would suggest toning down the mention of the joint modelling showed higher precision or smaller uncertainty throughout abstract, results and discussion. Indeed, the SEs between the two models are only marginally different.

Minor comments:

6. References #1 and #2 do not support the sentence they were cited for.

7, The link provided for data access does not provide a direct access to data. Suggest using this instead: https://extranet.who.int/ncdsmicrodata/index.php/catalog/633/.

8. Page 6 Line 133: please explain what "600 milliliters per kilogram" is referring to

9. Page 7 Line 141: please correct the sentence: were there two or three categories?

10. Page 7 Line 151: "or" should be "and"

11. Page 8 Line 186: how was complex survey design accounted for, eg through what command or package?

12. Page 9 equations: please simplify it to m = 2

13. Table 2: SE of effect on SBP for impaired FPG in joint model: the number 2.98 is likely a typo - it is the only number that is more than marginally different between the two models.

14. Table 2: I suggest using normal weight as the reference category for BMI.

7. PLOS authors have the option to publish the peer review history of their article (what does this mean?). If published, this will include your full peer review and any attached files.

**Do you want your identity to be public for this peer review?** For information about this choice, including consent withdrawal, please see our Privacy Policy.

Reviewer #1: **Yes: **Olivier Mukuku

Reviewer #3: No

---

## [Decision Letter · Decision Letter 2]

22 Aug 2024

Joint regression modeling of blood pressure and associated factors among adults in Uganda: implications for clinical practice

PGPH-D-24-00125R2

Dear Mr Appeli,

We are pleased to inform you that your manuscript 'Joint regression modeling of blood pressure and associated factors among adults in Uganda: implications for clinical practice' has been provisionally accepted for publication in PLOS Global Public Health.

Best regards,

Julia Robinson

Executive Editor

Reviewer Comments (if any, and for reference):

Reviewer's Responses to Questions

**Comments to the Author**

1. If the authors have adequately addressed your comments raised in a previous round of review and you feel that this manuscript is now acceptable for publication, you may indicate that here to bypass the “Comments to the Author” section, enter your conflict of interest statement in the “Confidential to Editor” section, and submit your "Accept" recommendation.

Reviewer #1: All comments have been addressed

2. Does this manuscript meet PLOS Global Public Health’s publication criteria? Is the manuscript technically sound, and do the data support the conclusions? The manuscript must describe methodologically and ethically rigorous research with conclusions that are appropriately drawn based on the data presented.

Reviewer #1: Yes

3. Has the statistical analysis been performed appropriately and rigorously?

Reviewer #1: Yes

4. Have the authors made all data underlying the findings in their manuscript fully available (please refer to the Data Availability Statement at the start of the manuscript PDF file)?

Reviewer #1: Yes

5. Is the manuscript presented in an intelligible fashion and written in standard English?

Reviewer #1: Yes

6. Review Comments to the Author

Reviewer #1: (No Response)

7. PLOS authors have the option to publish the peer review history of their article (what does this mean?). If published, this will include your full peer review and any attached files.

**Do you want your identity to be public for this peer review?** For information about this choice, including consent withdrawal, please see our Privacy Policy.

Reviewer #1: **Yes: **Olivier Mukuku
